# Societal Marketing in the Treatment of Type 2 Diabetes Mellitus: A Longitudinal Questionnaire Survey for Michelin-Starred Restaurants in Japan

**DOI:** 10.3390/ijerph16040636

**Published:** 2019-02-21

**Authors:** Satoru Yamada, Yoshifumi Yamada, Junichiro Irie, Kazuo Hara, Takashi Kadowaki, Yoshihito Atsumi

**Affiliations:** 1Kitasato Institute Hospital, Diabetes Center, 5-9-1 Shirokane, Minato-ku, Tokyo 108-8642, Japan; yamayoshi0718@yahoo.co.jp (Y.Y.); j-irie@z8.keio.jp (J.I.); 2Department of Medicine, Division of Endocrinology and Metabolism, Jichi Medical University, Saitama Medical Center, 1-847 Amanuma-cho, Omiya-ku, Saitama 330-8503, Japan; hara@jichi.ac.jp; 3Department of Prevention of Diabetes and Lifestyle-Related Diseases, Graduate School of Medicine, The University of Tokyo, 7-3-1 Hongo, Bunkyo-ku, Tokyo 113-0033, Japan; kadowaki-3im@h.u-tokyo.ac.jp; 4Department of Metabolism and Nutrition, Teikyo University, Faculty of Medicine, Mizonokuchi Hospital, Kanagawa 213-8507, Japan; 5Eiju General Hospital, Diabetes Center, 2-23-16 Higashi-Ueno, Taito-ku, Tokyo 110-8645, Japan; yatsumi@eijuhp.com

**Keywords:** type 2 diabetes mellitus, diet, food industry, restaurant

## Abstract

Numerous dietary recommendations have been made for the prevention and treatment of diabetes. However, many people with diabetes regard healthy dietary behavior as wearisome and have difficulty adhering to nutrition therapy. We, therefore, conducted a questionnaire survey concerning the restaurants that serve meals suitable for people with diabetes. We first aimed to determine the number of restaurants that were aware of the need to create special menus for people with diabetes. Second, we aimed to encourage restaurants’ serving of tasty, healthy food and promote easier social living for people with diabetes. We conducted our questionnaire survey every year from 2008 to 2013 on the availability of special menus for people with diabetes at restaurants listed in Michelin Guide Tokyo. We succeeded in increasing the proportion of restaurants offering special meals for people with diabetes from 6.7% (10 of 150 restaurants) in 2008 to 13.2% (32 of 242 restaurants) in 2013. As a result of the diabetes pandemic, the market for goods and services catering to people with diabetes is increasing. Diabetologists need to inform and support the food industry to produce foods that are suitable for people with diabetes and promote the serving of such foods by restaurants. This represents a new approach in the prevention and treatment of type 2 diabetes.

## 1. Introduction

Nutrition therapy is recommended for all people with diabetes [1]. Based on personal preferences, various eating patterns are acceptable for the management of diabetes. However, healthy dietary behavior is considered wearisome, and people with diabetes generally encounter difficulties in constantly adhering to their nutrition therapy. Hitherto, improving dietary patterns largely depends on the individual’s own efforts. Personal lifestyle choices and changes in the living environment beyond individual control may make treatment compliance easier for people with diabetes.

Neighborhood environments were found to be closely related to diabetes prevalence. The prevalence among Japanese-Americans living in Hawaii and the Los Angeles area was shown to be higher than that in Hiroshima [2]. Another study determined that a residential environment with greater resources for supporting physical activity and healthy diets was associated with a lower incidence of type 2 diabetes mellitus [3]. However, interventions directed at environmental factors, such as imposing a soda tax [4], have been rare, and few studies have involved medical professionals educating meal providers about offering special menus for people with diabetes. A variety of dietary options may make it easier for people with diabetes to adhere to nutrition therapy. We thought, however, that a change to the menu for people with diabetes in unknown restaurants or from a few food producers would not change the broader market (supermarket chain stores, convenience stores, online shops, fast-food chains, etc.). We estimated that altering the menu for people with diabetes in very famous restaurants could change such food service businesses. As a first step, we performed this study to engage such restaurants. Michelin Guide Tokyo [5,6,7,8,9,10] has been published annually since 2008, and the restaurants listed have gained large-scale media attention. We approached those restaurants to obtain their cooperation and hoped that their actions regarding nutrition therapy would have a spillover effect on Japan’s entire food industry.

In approaching the restaurants, we considered the following points: (1) Since chefs are busy individuals, the cooperative process had to be one that could be quickly completed; (2) Our approach would have to involve seeking the chefs’ opinions, not unilaterally imparting information from the standpoint of medical professionals. We thus adopted a simple questionnaire survey consisting of three questions about offering a specific menu for people with diabetes. We conducted that survey annually from 2008 to 2013. Many restaurant representatives completed the questionnaire more than once. The aim of this study is to elucidate whether the questionnaire items would induce the chefs to consider serving tasty, nutritious cuisine for people with diabetes and developing special menus for them.

## 2. Materials and Methods

We mailed the following questionnaire to restaurants listed in Michelin Guide Tokyo [5,6,7,8,9,10] every October from 2008 to 2013. Representatives of the restaurants completed and returned the questionnaire. The following questions required “yes” or “no” answers:Question 1: Do you offer a specific menu with items having either restricted calories (about 600 kcal) or reduced carbohydrates (less than 40 g) for people with diabetes?Question 2: Do you plan to develop a special menu for people with diabetes?Question 3: In the future, would you like to offer a special menu for people with diabetes?

Based on the questionnaire results and upon adopting the transtheoretical approach proposed by Prochaska et al. [11], we categorized the restaurants as follows: (1) Maintenance—restaurants that offered a special menu for people with diabetes; (2) Action—restaurants that were developing a special menu for people with diabetes; (3) Preparation—restaurants that were planning to develop such a menu; (4) Contemplation—restaurants with no special menu for people with diabetes and no concrete plans to develop one, but that were aiming to develop one in the future; (5) Precontemplation—restaurants with no special menu for people with diabetes that were not interested in developing one; and (6) No response—no answer, probably owing to indifference to customers’ health. 

If restaurants were planning to develop a specific menu for people with diabetes and requested assistance in doing so, we visited them to provide help. Following the dietary recommendations of the American Diabetes Association [1], we offered the following three diet options: Low-fat, calorie-restricted; Mediterranean; and low-carbohydrate (Table 1). We did not offer a vegetarian diet because most Michelin-starred restaurants preferred to offer meat and/or fish.

The frequencies of restaurants in each annual categorization were compared by a likelihood Chi-square test (Stata version 13.1, Stata Corp LLC, College Station, TX, USA).

## 3. Results

The number of restaurants in the maintenance, action, and preparation stages showed an annual increase during the study period, as indicated in Table 2. 

Notably, the number of restaurants offering a special menu showed a rise in 2012 (34/247 = 13.8%). Using a likelihood Chi-square test, there was a significant difference (*p* = 0.010) between 2008 (maintenance 10, other responses 40, unidentified 100) and 2012 (maintenance 34, other responses 83, unidentified 130). In 2013, however, the number of restaurants in the maintenance stage remained at roughly the same level as in 2012 (32/242 = 13.2%). Most restaurants selected the low-carbohydrate diet to provide a menu for diabetics (Table 3).

## 4. Discussion

Modifying neighborhood environments by increasing the availability of healthy foods and physical activity resources may help mitigate the risk for type 2 diabetes mellitus [3]. To enable the prevention and treatment of diabetes, we believe it would be valuable to raise public awareness regarding diabetes and obtain civil society and industry support. In this study, we targeted restaurants listed in Michelin Guide Tokyo as representative providers of quality food by means of a simple annual questionnaire survey. In that survey, we hoped Question 1 would increase chefs’ awareness of the situation regarding diabetes, Question 2 would induce them to consider appropriate recipes, and Question 3 would encourage them to develop a special menu in the future. Some chefs consulted us on how to develop a menu for people with diabetes. We proposed the following three dietary methods: Low-fat, calorie-restricted; Mediterranean; and low-carbohydrate. Most chefs chose the low-carbohydrate diet. As a general recommendation to reduce carbohydrate content, we suggested that bread and noodles be made from soybean flour or wheat bran instead of grain flour or other starchy ingredients. Concerning the low-carbohydrate diet, there was some debate. In fact, when the ADA first recommended a low-carbohydrate diet in 2008, they attached cautions about individuals’ renal function and lipid profiles [16]. However, the ADA excluded these cautions in 2013 [1]. Furthermore, a consensus report of the ADA/EASD in 2018 also recognized a low-carbohydrate diet as one of the recommendable diets with which no side effects were attached [17].

Through our questionnaire survey, we succeeded in getting 13% of the restaurants to offer special menus for diabetics during the five-year study period. Those special menus frequently appeared in magazines and on television. We expect that such special meals for people with diabetes will spread throughout the entire food-service industry in the near future. This will result in greater dietary compliance by people with diabetes and will, consequently, reduce their complications. It may be possible to facilitate social change by making meal providers conscious of healthy foods, rather than through legislative measures, such as a soda tax. We believe that our new approach can help prevent and treat type 2 diabetes. In fact, in Japan several convenience stores and fast-food chains started to serve low-carbohydrate foods.

This study has several limitations. First, we did not have a control group, which would have completed a different type of questionnaire or towards which we would have adopted a different approach. In this regard, it is necessary to consider which type of approach would be most effective. Second, we selected Michelin-starred restaurants for our intervention. Further research must establish whether similar results would be obtained from less popular restaurants.

There is a need for the establishment of an organization that could develop ordinary foods for people with diabetes. Such a body would ideally consist of suppliers of food ingredients (specialists in the nutritional value and taste of foods), food manufacturers (who prepare foods using the supplied ingredients), and marketing companies (concerned with the profitability of those products). The organization would have to hold regular meetings to share information and enable the exchange of opinions among the various companies involved. Finally, medical evaluation of the new products would be necessary, so as to guarantee their quality. Such products would have many benefits. Owing to the diabetes pandemic, the market for products targeted at people with diabetes is growing considerably, and the food industry can profit from sales in this area. People with diabetes would benefit in terms of both their health and eating pleasure. There would also be significant social gains through savings in treatment costs and reduced death and disability caused by preventable complications of diabetes.

It is our goal for such healthy foods to become commonly consumed over the world so that people with diabetes will be able to adhere to nutrition therapy and take delight in eating.

## 5. Conclusions

Through our longitudinal survey, it was revealed that the number of restaurants that provide menus for people with diabetes increased, and most of them involved low-carbohydrate diets. This increase in restaurants that provide menus for people with diabetes represents a social solution for dietary therapy in improving diabetes. In the near future, diabetologists need to inform and support the food industry concerning the production of foods that are suitable for people with diabetes and the promotion of offering such foods in restaurants and/or supermarkets. This represents a new approach in the prevention and treatment of type 2 diabetes.

## Figures and Tables

**Table 1 ijerph-16-00636-t001:** Summary of diet options.

Name [Reference]	Main Components
Low-carbohydrate [12]	Dietary patterns that restrict consumption of carbohydrates by increasing intake of fats and protein from animal or plant food sources
Low-fat, calorie-restricted [13]	Dietary patterns that restrict consumption of total energy to approximately 80% of the original intake, mainly by fat restriction.
Mediterranean [14]	High consumption of minimally processed plant-based foods; olive oil as the principal source of fat; low-to-moderate consumption of dairy products, fish, and poultry; low consumption of red meat; and low-to-moderate consumption of wine with meals
Vegetarian [15]	Dietary patterns that are devoid of all animal-derived products, with or without some exceptions (Vegan: no exception, Lacto-Ovo: consuming dairy or eggs, Semi: all but no red meat and poultry)

**Table 2 ijerph-16-00636-t002:** Annual categorization of restaurants based on the survey responses.

	2008	2009	2010	2011	2012	2013
Maintenance	10 (6.7)	2 (1.2)	7 (3.6)	7 (2.9)	34 (13.8)	32 (13.2)
Action	0 (0.0)	0 (0.0)	1 (0.5)	7 (2.9)	2 (0.8)	0 (0.0)
Preparation	3 (2.0)	7 (4.0)	14 (7.1)	10 (4.2)	6 (2.4)	0 (0.0)
Contemplation	4 (2.7)	5 (2.9)	27 (13.7)	21 (8.8)	27 (10.9)	19 (7.9)
Precontemplation	33 (22.0)	39 (22.5)	19 (9.6)	42 (17.5)	48 (19.4)	30 (12.4)
No response	100 (66.7)	120 (69.4)	129 (65.5)	153 (63.8)	130 (52.6)	161 (66.5)
Total	150 (100)	173 (100)	197 (100)	240 (100)	247 (100)	242 (100)

The numbers represent the number(percentage) of participating restaurants in each of year of study.

**Table 3 ijerph-16-00636-t003:** Change in the restaurants’ attitudes regarding specific menus for people with diabetes.

	2008	2009	2010	2011	2012	2013
Low-carb	0	0	5	6	34	32
Low-fat, calorie-restricted	1	0	0	0	0	0
Mediterranean	0	0	0	0	0	0
Vegetarian	1	0	0	0	0	0
Others (Unknown)	8	2	2	1	0	0
Total	10	2	7	7	34	32

The numbers represent the number of restaurants adhering to each type of diet.

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
