# Peer review of "Societal Marketing in the Treatment of Type 2 Diabetes Mellitus: A Longitudinal Questionnaire Survey for Michelin-Starred Restaurants in Japan"

_ijerph, 2019, doi:10.3390/ijerph16040636_

Round 1

Reviewer 1 Report

The paper of Yamada et al. describe the use of the instrument of social marketing in the treatment of type 2 diabetes mellitus. They report the outcome of a longitudinal questionaire survey for Michelin’s starred restaurants. They claim that this represents a new approach in the prevention and treatment of type 2 diabetes (line33).

The reviewer supports the idea that promoting nutritional therapy of type 2 diabetes is highly relevant.

Major comments: This manuscript is lacking a clear research question.The authors do not motivate their choices of instruments: the role of social marketing vs other instruments, their choice of restaurant menu’s, their choice of Michelin starred restaurants.

A combination with an overall advertisement campaign and measuring the impact in the restaurant’s attitude would be interesting.

The contribution of dining  (sometimes?) in a Michelin starred restaurant  to the health of an individual diabetic patient can be neglected, but this might not be the aim.

Regarding to the aspects of the diets: the role of vegetables and fish is not mentioned  

Low carb diets seem to be the most important issue in the results. This might be the reflection of a broad change in the consumers opinion in the last years. However, this is highly debatable: it does not take into consideration that slow carbs and dietary fiber contibute to the health of diabetic patients.

Author Response

Responses to the comments from Reviewer #1

1.      The reviewer supports the idea that promoting nutritional therapy of type 2 diabetes is highly relevant.

Response: We greatly appreciate your encouraging comment.

2.      This manuscript is lacking a clear research question.

The authors do not motivate their choices of instruments: the role of social marketing vs other instruments, their choice of restaurants menu's, their choice of Michelin starred restaurants.

Response: We appreciate your comment. Our research questions are as follows:

(1)   Will a questionnaire survey about a menu for people with diabetes help to develop a menu for people with diabetes in Michelin-starred restaurants? (current manuscript)

(2)   Will the development of a menu for people with diabetes in Michelin-starred restaurants help to develop food products for people with diabetes in the broader market (chain stores of supermarkets and CVS, on-line shops, fast-food chains, etc.)? (on-going project)

(3)   Will the development of food products in the broader Japanese market help to develop food products for people with diabetes all over the world after the Olympic Games in Tokyo in 2020? (future approach)

We also made the following changes to the sentence shown below:

The aim of this study is to elucidate we expected that whether the questionnaire items would induce the chefs to consider serving tasty, nutritious cuisine for diabetics and developing special menus for them.” (P2, lines 64)

3.      A combination with an overall advertisement campaign and measuring the impact in the resutaurant's attitude would be interesting.

Response: We completely agree with your opinion. We will consider this approach in the future.

4.      Regarding to the aspects of the diets: the role of vegetables and fish is not mentioned.

Response: Although we added Table 1 in the “Materials and Methods” section to describe several dietary approaches for people with diabetes, the Michelin-starred restaurants that responded to our questionnaire served meat combinations with vegetables and fish. Thus, we did not mention the specific roles of vegetables and fish in this manuscript.

Furthermore, we moved Table 2 (Table 1 in previous manuscript) from “Materials and Methods” section to “Results” section.

5.      Low carb diets seem to be the most important issue in the results.

This might be the reflection of a broad change in the consumers opinion in the last years.

However, this is highly debatable: it does not take into consideration that slow carbs and dietary fiber contibute to the health of diabetic patients.

Response: We appreciate your suggestion. As mentioned above, we added Table 1 to explain dietary approaches including a low-carbohydrate diet and added the following sentences in the “Discussion” section.

“Concerning the low-carbohydrate diet, there was some debate. In fact, when the ADA first recommended a low-carbohydrate diet in 2008, they attached cautions about individuals’ renal function and lipid profiles [16]. However, the ADA excluded these cautions in 2013 [1]. Furthermore, a consensus report of the ADA/EASD in 2018 also recognized a low-carbohydrate diet as one of the recommendable diets with which no side effects were attached [17]” (p 4, lines 127-131).

                Furthermore, we added the following references:

12.    Accurso, A.; Bernstein, R.K.; Dahlqvist, A.; Draznin, B.; Feinman, R.D.; Fine, E.J.; Gleed, A.; Jacobs, D.B.; Larson, G.; Lustig, R.H.; et al. Dietary carbohydrate restriction in type 2 diabetes mellitus and metabolic syndrome: Time for a critical appraisal. Nutr Metab (Lond) 2008, 5(9), DOI: 10.1186/1743-7075-5-9.

13.    Tajima, N.; Noda, M.; Origasa, H.; Noto, H.; Yabe, D.; Fujita, Y.; Goto, A.; Fujimoto, K.; Sakamoto, M.; Haneda, M. Evidence-based practice guideline for the treatment for diabetes in Japan 2013. Diabetol Int 2015, 6, 151-187. DOI: 10.1007/s13340-015-0206-2.

14.    Esposito, K.; Maiorino, M.I.; Ceriello, A.; Giugliano, D. Prevention and control of type 2 diabetes by Mediterranean diet: A systematic review. Diabetes Res Clin Pract 2010, 89, 97-102. DOI: 10.1016/j.diabres.2010.04.019.

15.    Tonstad, S.; Stewart, K.; Oda, K.; Batech, M.; Herring, R.P.; Fraser, G.E. Vegetarian diets and incidence of diabetes in the Adventist Health Study-2. Nutr Metab Cardiovasc Dis 2013, 23, 292-299. DOI: 10.1016/j.numecd.2011.07.004.

16.    American Diabetes Association. Nutrition recommendations and interventions for diabetes: A position statement of the American Diabetes Association. Diabetes Care 2008, 31(suppl 1), S61-78. DOI: 10.2337/dc08-S061.

17.    Davies, M.J.; D’Alessio, D.A.; Fradkin, J.; Kernan, W.N.; Mathier, C.; Mingrone, G.; Rossing, P.; Tsapas, A.; Wexler, D.J.; Buse, J.B. Management of hyperglycemia in type 2 diabetes, 2018. A consensus report by the American Diabetes Association (ADA) and the European Association for the Study of Diabetes (EASD). Diabetes Care 2018, 41, 2669-2701. DOI: 10.2337/dci18-0033.

Reviewer 2 Report

a greater description of the three proposed diets would be useful

the paper  is quite original and the aim of promoting healthy eating is worthwhile. The work as a whole could be published as a note as it does not take up much space. The description or reference of the proposed diets would have been useful.

Author Response

Responses to the comments from Reviewer #2

1.      A greater description of the three proposed diets would be useful.

The description or reference of the proposed diets would have been useful.

Response: We appreciate your suggestion. We added Table 1 in the “Materials and Methods” section. Further, we added 4 references.

12.    Accurso, A.; Bernstein, R.K.; Dahlqvist, A.; Draznin, B.; Feinman, R.D.; Fine, E.J.; Gleed, A.; Jacobs, D.B.; Larson, G.; Lustig, R.H.; et al. Dietary carbohydrate restriction in type 2 diabetes mellitus and metabolic syndrome: Time for a critical appraisal. Nutr Metab (Lond) 2008, 5(9), DOI: 10.1186/1743-7075-5-9.

13.    Tajima, N.; Noda, M.; Origasa, H.; Noto, H.; Yabe, D.; Fujita, Y.; Goto, A.; Fujimoto, K.; Sakamoto, M.; Haneda, M. Evidence-based practice guideline for the treatment for diabetes in Japan 2013. Diabetol Int 2015, 6, 151-187. DOI: 10.1007/s13340-015-0206-2.

14.    Esposito, K.; Maiorino, M.I.; Ceriello, A.; Giugliano, D. Prevention and control of type 2 diabetes by Mediterranean diet: A systematic review. Diabetes Res Clin Pract 2010, 89, 97-102. DOI: 10.1016/j.diabres.2010.04.019.

15.    Tonstad, S.; Stewart, K.; Oda, K.; Batech, M.; Herring, R.P.; Fraser, G.E. Vegetarian diets and incidence of diabetes in the Adventist Health Study-2. Nutr Metab Cardiovasc Dis 2013, 23, 292-299. DOI: 10.1016/j.numecd.2011.07.004.

Reviewer 3 Report

I find the idea of implementing a menu for diabetics interesting.

The increase in the number of restaurants offering a special menu is therefore encouraging.

The potential positive effect on population health remains though unclear.

Major comments:

It would be interesting to see if the attitude of specific restaurants changes throughout the years. I believe some restaurants are mentioned in the guide year after year. Could you run an analysis on these selected restaurants?

In the western civilization, Diabetes is more prevalent among low socio-economic populations. I wonder whether implementation of menus for diabetics make more sense in fast-food chains rather than in restaurants recommended by the Michelin guide. Please discuss.

Are there any studies showing that offering healthier foods in restaurants or cafeterias reduce with time rates of diabetes or cardiovascular diseases? The idea behind the study is nice, but the question is whether the customers are compliant and always order these menus. How often? Will they return home and eat unhealthy again? Is this approach effective in the long-run? Please discuss.

Minor comments:

Did the authors actually visit the restaurants or only mailed the survey? This study may assume then that the restaurants are compliant. There could be cases that although they say they offer a menu for diabetics, in practice they don’t. Their answers require verification. Assessing their menus online could be sufficient as well.

Line 77- 6) unidentified – you cannot conclude that restaurants are indifferent to customers’ health. It’s possible they are just non-cooperative.

Author Response

Responses to the comments from Reviewer #3

1.      It would be interesting to see if the attitude of specific restaurants changes throughout the years.

I believe some restaurants are mentioned in the guide year after year.

Could you run an analysis on these selected restaurants?

Response; We appreciate your interesting point of view. Among 150 restaurants in Michelin Tokyo 2008, only 54 restaurants survived in Michelin Tokyo 2013. In these 54 restaurants, 8 restaurants served low-carbohydrate diet in our survey. Eight of 54 (14.8%) seems almost similar with other restaurants in Michelin Tokyo 2013 (24/188 = 12.8%).

2.      In the western civilization, diabetes is more prevalent among low socio-economic populations.

I wonder whether implementation of menus for diabetics make more sense in fast-food chains rather than in restaurants recommended by the Michelin guide.

Please discuss.

Response: We agree with you completely. In Japan, the CVS chain “Lawson” started to sell “bran bread,” which is a low-carbohydrate bread, in 2012 (http://lawson.jp/en/product_service/product/). In addition, the largest fast-food chain, “Zensho” has had a low-carbohydrate menu since 2017 (Please see the text in the attached pdf on page 2 highlighted in yellow.) Furthermore, we added the following sentence in “Discussion” section:

“In fact, in Japan several CVS and fast-food chains have started to serve low-carbohydrate foods.” (page 4, lines 139-140)

3.      Are there any studies showing that offering healthier foods in restaurants or cafeterias reduce with time rates of diabetes or cardiovascular diseases?

Response We appreciate for you important comment. We think our study is the first trial to evaluate the value of societal marketing in the treatment and prevention of type 2 diabetes. We have never seen any other such studies.

4.      The idea behind the study is nice, but the question is whether the customers are compliant and always order these menus.

How often?

Will they return home and eat unhealthy again?

Response: Once again, we appreciate your thoughtful comments. We think visiting Michelin-starred restaurants is not something the general population does very often. However, we believe the food culture will expand from Michelin-starred restaurants to CVS and fast-food chains within a short period of time. In such a situation, people with diabetes will be able to continue eating healthily in various life situations.

5.      Is this approache effective in the long-run?

Please discuss.

Response: We think long-term feasibility is very important in dietary approaches, as you pointed out. Because we stopped this survey in 2013, according to the study design, we cannot judge whether this approach is effective in the long-run. However, many restaurants with or without Michelin Star ratings have started to serve healthier menus in Kobe (http://locabo-kobe.com/), Maruouchi (Tokyo) (https://shokumaru.jp/20170627_1/) and Kashiwa (https://www.facebook.com/events/245673839325389/). Although these restaurants are not related with current study, first author (SY) of this study supports them. This suggests the long-term feasibility of our project.

6.      Did the authors actually visit the restaurants or only mailed the survey?

Response: We visited the restaurants one-by-one to discuss the suitability of the menus according to chefs’ preferences.

7.      This study may assume then that the restaurants are compliant. These could be cases that although they say they offer a menu for diabetics, in practice they don't.

Their answers require verification.

Response: Certainly, we might assume that the chefs are merely being compliant. However, most of the Michelin-starred chefs felt “Noblesse Oblige.” Thus, when we visited their restaurants with expectations of a healthier menu, they served a perfect menu.

8.      Assessing their menus online could be sufficient as well.

Response; Most of Michelin starred Chefs dislike online business. They like over the counter serving. Thus, we cannot perform such a study.

9.      Line77-6) unidentified- you cannot conclude that restaurants are indifferent to customers' health.

It's possible they are just non-cooperative.

Response; We agree with you. Thus, we changed the term “unidentified” to “no response” in Table 2. We have also slightly modified the statement regarding restaurants being indifferent to customers’ health.

Round 2

Reviewer 1 Report

As  indicated before I questioned the methodology of the study (Why high-quaity restaurants only relevant to the upperclass?) and the relevance. 

The revised version only contains more references and does not address my major issues

Author Response

Response to the comment from Reviewer #1

As indicated before, I questioned the methodology of the study (Why high-quality restaurants only relevant to the upper-class?) and the relevance. 

Response: We might have misunderstood the intention of the previous comment from Reviewer 1. The reasons why we chose the Michelin-starred restaurants (our hypothesis) are shown below:

1.      The chefs in Michelin-starred restaurants will never cook below-average meals in terms of taste.

2.      For that reason, meals in Michelin-starred restaurants will satisfy many people, with or without diabetes.

3.      Thus, the menu for people with diabetes in Michelin-starred restaurants will satisfy people with diabetes and help to secure the sustainability of such meals in restaurants.

4.      The implementation of menus for people with diabetes in Michelin-starred restaurants will create the perception that there is a substantial need and an untapped market in the industry for delicious foods for people with diabetes.

5.      After the development of a menu for people with diabetes in Michelin-starred restaurants, the food industry, including fast-food chains, will likely develop products for people with diabetes.

To clarify our purpose, we made the following changes to the “INTRODUCTION” section:

In this study, we engaged with restaurants and other food service businesses toward that end. We thought, however, that a change to the menu for people with diabetes in unknown restaurants or from a few food producers would not change the broader market (supermarket chain stores, convenience stores, on-line shops, fast-food chains, etc.). We estimated that altering the menu for people with diabetes in very famous restaurants could change such food service businesses. As a first step, we performed this study to engage such restaurants” (p. 2, lines 53-59).

We believe that these sentences will help Reviewer 1 to better understand our methodology.